

# Reporting and methodological quality of meta-analyses in urological literature

Leilei Xia[1], Jing Xu[2] and Thomas J. Guzzo[1]

[1] Division of Urology, Department of Surgery, University of Pennsylvania Perelman School of Medicine, Philadelphia, PA, United States
[2] Cerebral Vascular Disease Research Laboratories, Department of Neurology and Neuroscience Program, University of Miami Miller School of Medicine, Miami, FL, United States

## ABSTRACT

**Purpose**. To assess the overall quality of published urological meta-analyses and identify predictive factors for high quality.

**Materials and Methods**. We systematically searched PubMed to identify meta-analyses published from January 1st, 2011 to December 31st, 2015 in 10 predetermined major paper-based urology journals. The characteristics of the included meta-analyses were collected, and their reporting and methodological qualities were assessed by the PRISMA checklist (27 items) and AMSTAR tool (11 items), respectively. Descriptive statistics were used for individual items as a measure of overall compliance, and PRISMA and AMSTAR scores were calculated as the sum of adequately reported domains. Logistic regression was used to identify predictive factors for high qualities.

**Results**. A total of 183 meta-analyses were included. The mean PRISMA and AMSTAR scores were $22.74 \pm 2.04$ and $7.57 \pm 1.41$, respectively. PRISMA item 5, protocol and registration, items 15 and 22, risk of bias across studies, items 16 and 23, additional analysis had less than 50% adherence. AMSTAR item 1, "*a priori*" design, item 5, list of studies and item 10, publication bias had less than 50% adherence. Logistic regression analyses showed that funding support and "*a priori*" design were associated with superior reporting quality, following PRISMA guideline and "*a priori*" design were associated with superior methodological quality.

**Conclusions**. Reporting and methodological qualities of recently published meta-analyses in major paper-based urology journals are generally good. Further improvement could potentially be achieved by strictly adhering to PRISMA guideline and having "*a priori*" protocol.

Corresponding authors
Leilei Xia, leilei.xia@uphs.upenn.edu
Thomas J. Guzzo,
thomas.guzzo@uphs.upenn.edu

## INTRODUCTION

A systematic review is a review of a clearly formulated question using systematic methods to identify, select and critically appraise relevant research. The systematic review may include a quantitative synthesis of results called meta-analysis, which summarizes all results of primary studies in order to obtain a combined estimate of the effect. Certain types of systematic review and meta-analysis are considered as the highest level of evidence (level 1a) (http://www.cebm.net/oxford-centre-evidence-based-medicine-levels-evidence-march-2009/). Also, well-conducted meta-analyses can sometimes resolve conflicting evidence and provide

more reliable conclusions (*Berlin & Golub, 2014*). Meta-analyses are often appealing to both authors and journals as they are commonly highly cited publication. Rapidly expanding literature across all medical disciplines raise the increasing need to summarize and synthesis the currently available evidence. Such factors have contributed to the increased number of published meta-analyses in medical journals (*Tunis et al., 2013*; *Zhang et al., 2016*).

However, like original research articles, quantity does not mean quality (*Adie et al., 2015*; *Berlin & Golub, 2014*; *Dechartres et al., 2014*; *Murad & Montori, 2013*). It is imperative for both the medical and publishing community to aware the negative influence of flawed or low-quality meta-analyses (*Berlin & Golub, 2014*). Several statements or guidelines have been proposed and validated as the tools to assess the quality of published meta-analyses (*Faggion, 2015*; *Liberati et al., 2009*; *Moher et al., 2009*; *Pieper et al., 2015*; *Shea et al., 2007a*; *Shea et al., 2007b*; *Shea et al., 2009*; *Stroup et al., 2000*). The most well-known guideline is the Preferred Reporting Items for Systematic Reviews and Meta-Analyses (PRISMA), which is actually a checklist recommended to follow when reporting meta-analyses (*Liberati et al., 2009*; *Moher et al., 2009*). An earlier initiative was the development of the AMSTAR, a measuring tool to assess the methodological or conducting quality of meta-analyses (*Shea et al., 2007b*; *Shea et al., 2009*). In other words, AMSTAR usually serves as a critical appraisal tool to identify the scope of bias in methodology at the review level.

Although still debatable, a number of studies in various surgical and medical fields have used the "scores" based on the fulfillments of PRISMA and AMSTA to assess the qualities of systematic reviews and meta-analyses (*Adie et al., 2015*; *Gagnier & Kellam, 2013*; *Liu et al., 2017*; *Shea et al., 2007a*; *Tunis et al., 2013*; *Zhang et al., 2016*). There exist several duplicate items in the two tools, generally they are considered separate tools and are commonly used together for the assessment, PRISMA for reporting quality and AMSTAR for methodological quality (*Adie et al., 2015*; *Gagnier & Kellam, 2013*; *Liu et al., 2017*; *Shea et al., 2007a*; *Tunis et al., 2013*; *Zhang et al., 2016*).

To date, no studies have comprehensively assessed the reporting and methodological quality of urological meta-analyses, in particular those published after the PRISMA initiative (2009). Considering meta-analyses are often influential and highly cited publications, there is a need to explore whether general characteristics (author, journal, and report) of the meta-analyses have an association with the overall quality. Therefore, in the present study, we specifically focused on meta-analyses published in major paper-based urology journals, with the aim to assess the reporting and methodological quality, as well as to identify relevant predictive or associated factors.

## METHODS

### Eligibility criteria

To be eligible for inclusion, a meta-analysis had to meet the following inclusion criteria: (1) a study with the meta-analytic methodology pooling results from primary articles (including meta-analysis alone or systematic reviews containing meta-analyses); (2) published in the following 10 predetermined urology journals: British Journal of Urology International (BJUI), European Urology (EU), Journal of Endourology (JEU), Journal of Pediatric Urology (JPU), Journal of Urology (JU), Neurourology and Urodynamics (NUUD), Urology

(URO), Urolithiasis (UL), formerly known as Urological Research, Urologic Oncology (UO) and World Journal of Urology (WJU); (3) published in the printed journal between January 1st 2011 to December 31th 2015 (excluding "Epub ahead of print").

Exclusion criteria were: (1) systematic review without meta-analysis; (2) original research article or original research article combined with a meta-analysis; (3) network meta-analysis or multiple group comparison meta-analysis; (4) meta-analysis of single proportions; (5) meta-analysis originally published in the Cochrane Database of Systematic Reviews. There were two reasons for our decision to exclude the network meta-analysis (multiple group comparison meta-analysis) and meta-analysis of single proportions. First, they were relatively uncommon compared to other "traditional" meta-analyses. Second, the methods and results reported by those meta-analyses were very heterogenic and different from "traditional" or pairwise meta-analyses (*Bafeta et al., 2013*; *Bafeta et al., 2014*). Some of the items in the PRISMA and AMSTAR do not perfectly apply to network meta-analyses and meta-analysis of single proportions.

Two investigators (LX, JX) independently screened the titles and abstracts of all the identified references. Full-text were then retrieved for potential eligible meta-analysis. Discrepancies were resolved by discussion between the two investigators.

## Search strategy

The objective was to identify all the meta-analyses published from January 1st, 2011 to December 31st, 2015 in 10 predetermined major paper-based urology journals: BJUI, EU, JEU, JPU, JU, NUUD, URO, UL, UO, and WJU. We performed a focused search on PubMed. For BJUI, the search strategy was: *("BJU international"[Journal]) AND (meta-analysis [Title/Abstract] OR systematic review [Title/Abstract]) AND ("2011/01/01"[Date—Publication]: "2015/12/31"[Date—Publication]).* For other journals, the search strategy was the same except for the journal name.

## Data extraction

We collected all data on general characteristics of the included meta-analyses, and the key reporting (PRISMA) and methodological (AMSTAR) components of the meta-analysis process. Two investigators (LX, JX) independently extracted the data. Any disagreements were resolved by discussion between the two investigators. Inter-observer reliability was examined using the kappa ($\kappa$) value.

## General characteristics

We collected data on the following general characteristics: (1) corresponding author's region and country; (2) number of authors; (3) presence or absence of a professional with the background of epidemiology or statistics as a coauthor (including the acknowledgement part); (4) number of participating centers (department level); (5) subspecialties in urology (based on the American Urological Association classification); (6) presence or absence of any funding source; (7) the number of included studies; (8) type of the included studies (only RCTs or RCTs plus non-RCTs or only non-RCTs); (9) type of the meta-analyses (interventional, diagnostic, incidence related, prognostic or cannot classify); (10) type of the interventional meta-analyses (surgical or non-surgical); (11) attached a PRISMA checklist

or not; (12) followed the PRISMA guideline or not (claimed this in the article or not); (13) provided the protocol and registration information or not, which also referred to the PRISMA item 5; (14) "*a priori*" design or not (claimed this in the article or not), which also referred to the AMSTAR item 1.

## Assessment of key reporting components in the meta-analysis process

The PRISMA statement is a checklist of 27 items that are recommended to be included in systematic review and meta-analysis to ensure that the published report contains all relevant information (Supplemental Information). The present study focused only on meta-analyses and every item was applicable. Each PRISMA item was rated with a "yes" or "no" response. A "yes" response means that the item was reported, and a "no" response means that the item was not reported. For the purpose of data analysis, reported points were assigned as follows: "yes" = 1, "no" = 0. Therefore every included meta-analysis had an overall PRISMA score rated out of a maximum score of 27.

## Assessment of key methodological components in the meta-analysis process

The AMSTAR tool is an 11-item questionnaire that was used to determine the methodological or conducting quality of systematic reviews and meta-analyses (Supplemental Information). The original tool had four responses with each item, "yes," "no," "can't answer," or "not applicable." Due to the fact that we focused on meta-analyses (excluded pure systematic reviews), every item was applicable. Each AMSTAR item was rated with a "yes" "no" or "cannot answer" response. A "yes" response means that the item is fulfilled, a "no" response means that the item is not fulfilled, a "can't answer" response means that it is inconclusive as to whether the item is fulfilled. For the purpose of data analysis, reported points were assigned as follows: "yes" = 1, "no" or "can't answer" = 0. Therefore, every included meta-analysis had an overall AMSTAR score rated out of a maximum score of 11.

## Data analysis

Analyses, tables, and figures were configured by using a spreadsheet program (Excel 2013, Microsoft) and a statistical software (STATA 14.0, StataCorp LP). A descriptive analysis was performed for PRISMA and AMSTAR scores grouped by multiple categories. Shapiro–Wilk test was used to assess the normality of the PRISMA and AMSTAR scores ($p = 0.376$ and $p = 0.057$, respectively). Based on the distributions of PRISMA and AMSTAR scores and Shapiro–Wilk test results, we used the parametric tests to compare the qualities. Comparisons of mean qualities between dichotomous factors were conducted using the independent Student's $t$-test. Comparisons of study qualities between multifactor variables were conducted using the one-way analysis of variance (ANOVA), with the Tukey's HSD *post hoc* test. The PRISMA score and AMSTAR score were both divided into the superior and inferior quality groups with a cutoff value of 75% percentile of the respective ranges. Univariate logistic regression analysis was used to compare the differences between the superior and inferior groups with potential factors affecting study qualities. Variables included continent origin, country origin, number of authors, presence or absence of a professional with the
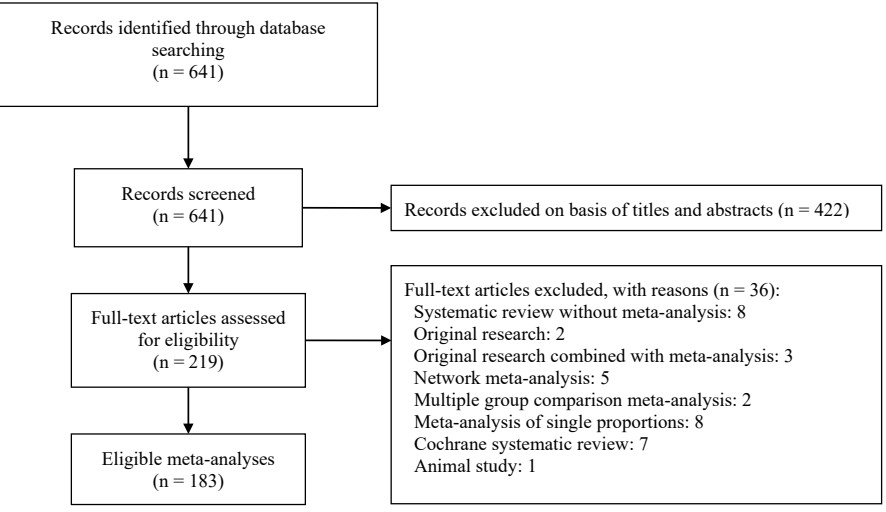

**Figure 1  Flow diagram of included meta-analyses.**

background of epidemiology or statistics as a coauthor, number of participating centers, subspecialties, funding support, number of included studies, type of the included studies, interventional meta-analysis, type of interventional meta-analysis, followed the PRISMA guideline, and ''*a priori*'' design. Factors that found to be significant ($p < 0.1$) were then entered into the multivariate logistic regression analysis. A $p < 0.05$ was considered significant on statistical analyses. All the analyses were two-sided.

# RESULTS

## Search results

Figure 1 depicts a flow diagram of meta-analyses selection. The initial search identified 641 references with potential relevance. Screening the title and abstract excluded 422 references and another 36 references were excluded after reviewing the full-text. Finally, 183 meta-analyses were included for the final assessment and analysis (Supplemental Information).

## General characteristics of the meta-analyses

The characteristics of the 183 meta-analyses are shown in Table 1. The number of authors and number of included studies were divided into two groups with the cutoff setting at median values (7 and 10, respectively). The number of patients included per meta-analysis ranged from 152 to 4082606. The number of patients in 14 studies could not be determined. EU ($n = 44$, 24%) published the largest number of included meta-analyses and JPU ($n = 2$, 1%) had the lowest number. The region where the largest number of included meta-analyses originated was Asia ($n = 89$, 49%). The most common countries of publication were China ($n = 82$, 45%), the USA ($n = 22$, 12%), and the UK ($n = 20$, 11%). Forty-four (24%) meta-analyses had at least one professional with the background of epidemiology or statistics as the coauthor (including the acknowledgement part). The most common subspecialty of the included meta-analyses was urologic oncology ($n = 75$, 41%). Most of the included meta-analyses could be categorized as interventional ($n = 141$,

77%), which were further subcategorized as surgical intervention (96/141, 68%) and non-surgical intervention (45/141, 32%). Sixty-two (34%) meta-analyses included only RCTs, 71 (39%) included only non-RCTs, and another 50 (27%) included both RCTs and non-RCTs. Fifty-five (30%) meta-analyses received funding support. Only two (1%) meta-analyses attached the PRISMA checklist. Fifty-five (30%) meta-analyses claimed followed the PRISMA guideline. Eight (4%) meta-analyses provided the protocol and registration information (PRISMA item 5) and 22 meta-analyses claimed the "*a priori*" design (AMSTAR item 1).

## Reporting quality (PRISMA)

The overall mean PRISMA score of all the included meta-analyses was $22.74 \pm 2.04$ (84.2% of items adequately reported, on average). Most of the PRISMA items (25 out of 27) had a $\kappa$ value more than 0.65 and none of them had a $\kappa$ value less than 0.5. Table 1 shows mean PRISMA scores grouped with various factors. After excluding journals (JPU, NUUD, and UO) with less than 10 included meta-analyses, EU had the highest mean PRISMA score ($23.52 \pm 1.93$). However, one-way ANOVA of PRISMA score showed no significant difference between the 7 journals (BJUI, EU, JEU, JU, URO, WJU, and UL), $F(6, 165) = 1.71, p = 0.12$. Student's $t$-test showed no significant difference of PRISMA score between the meta-analyses from Asia and those from non-Asia region, $t(181) = -0.50, p = 0.62$. There was no significant difference in PRISMA scores between the meta-analyses from China and remaining countries, $t(181) = -0.15, p = 0.88$. Included meta-analyses in the subspecialty of urologic oncology had higher PRISMA scores than other specialties, $t(181) = -2.09, p = 0.037$. There was no significant difference in PRISMA scores between the included studies type (Only RCT vs. RCT & non-RCT vs. Only non-RCT), $F(2, 180) = 0.47, p = 0.63$. There was no significant difference in PRISMA scores between the interventional and non-interventional meta-analyses, $t(181) = 1.80, p = 0.07$. All other two-group comparison test results are shown in Table 1.

Figure 2A shows the PRISMA results on a per-item basis. Per-item PRISMA analysis revealed that five items had less than 50% adherence out of the 183 included meta-analyses (item 5, protocol and registration; items 15 and 22, risk of bias across studies; items 16 and 23, additional analysis). Item 8 (search) also had only 51% adherence.

## Methodological quality (AMSTAR)

The overall mean AMSTAR score of all the included meta-analyses was $7.57 \pm 1.41$ (68.8% of items adequately reported, on average). Most of the AMSTAR items (10 out of 11) had a $\kappa$ value more than 0.65 and none of them had a $\kappa$ value less than 0.5. Table 1 shows mean AMSTAR scores grouped with various factors. After excluding journals (JPU, NUUD, and UO) with less than 10 included meta-analyses, EU had the highest mean AMSTAR score ($7.98 \pm 1.47$). One-way ANOVA of AMSTAR score showed significant differences between the 7 journals (BJUI, EU, JEU, JU, URO, WJU, and UL), $F(6, 165) = 3.03, p = 0.008$. Tukey's HSD *post hoc* test only showed that EU had higher AMSTAR score than URO, $p = 0.034$. Student's $t$-test showed no significant difference in AMSTAR score between the meta-analyses from Asia and those from non-Asia region, $t(181) = 0.06, p = 0.95$. There was no

**Table 1 Characteristics of included meta-analyses and quality scores assessed by the PRISMA checklist and AMSTAR tool.**

| Items | N (%) | PRISMA, mean (SD) | AMSTAR, mean (SD) |
|---|---|---|---|
| **Journal** | | | |
| BJUI | 27 (15%) | 22.85 (2.13) | 7.93 (1.14) |
| EU | 44 (24%) | 23.52 (1.93) | 7.98 (1.47) |
| JEU | 21 (11%) | 22.62 (1.53) | 7.95 (1.02) |
| JPU | 2 (1%) | 23 (1.41) | 9 (1.41) |
| JU | 25 (14%) | 22.52 (2.37) | 7.2 (1.68) |
| NUUD | 3 (2%) | 21.67 (0.58) | 7.33 (1.15) |
| U | 25 (14%) | 22.2 (1.61) | 6.92 (1.15) |
| UL | 18 (10%) | 22.67 (2.22) | 7.44 (1.24) |
| UO | 6 (3%) | 22.16 (3.18) | 7.17 (1.94) |
| WJU | 12 (7%) | 22.08 (2.23) | 6.92 (1.56) |
| **Origin region** | | | |
| Asia | 89 (49%) | 22.82 (1.94) | 7.56 (1.28) |
| Europe | 54 (30%) | 22.78 (2.05) | 7.80 (1.42) |
| North America | 29 (16%) | 22.66 (2.22) | 7.41 (1.59) |
| Oceania | 8 (4%) | 22.13 (2.9) | 7.13 (2.10) |
| South America | 3 (2%) | 22.33 (1.53) | 6.33 (0.58) |
| **Origin country** | | | |
| Australia | 6 (3%) | 22.33 (3.01) | 7.17 (2.14) |
| Austria | 3 (2%) | 22.67 (3.06) | 7 (1.73) |
| Brazil | 4 (2%) | 21.5 (2.08) | 6 (.82) |
| Canada | 7 (4%) | 22.29 (1.98) | 7.57 (1.51) |
| China[a] | 82 (45%) | 22.77 (1.95) | 7.60 (1.26) |
| France | 3 (2%) | 22.67 (1.53) | 8 (1) |
| Italy | 17 (9%) | 21.59 (2.06) | 7.24 (1.71) |
| The Netherlands | 7 (4%) | 24 (1.73) | 7.86 (0.69) |
| UK | 20 (11%) | 23.55 (1.682) | 8.6 (0.94) |
| USA | 22 (12%) | 22.77 (2.33) | 7.36 (1.65) |
| Other Countries | 12 (7%) | 23 (1.65) | 7.08 (1.51) |
| **Number of authors** | | | |
| ≥7 | 93 (51%) | 22.86 (1.92) | 7.72 (1.42) |
| <7 | 90 (49%) | 22.62 (2.16) | 7.41 (1.40) |
| **Statistician as coauthor** | | | |
| Yes (1) | 44 (24%) | 22.68 (2.45) | 7.48 (1.64) |
| No (0) | 139 (76%) | 22.76 (1.91) | 7.60 (1.34) |
| **Participating center** | | | |
| Single | 46 (25%) | 22.57 (1.76) | 7.48 (1.21) |
| Multiple | 137 (75%) | 22.80 (2.13) | 7.60 (1.48) |
| **Subspecialty** | | | |
| Pediatric urology | 13 (7%) | 21.77 (2.59) | 7.15 (2.03) |
| Urologic oncology | 75 (41%) | 23.12 (2.16) | 7.65 (1.50) |

**Table 1** (*continued*)

| Items | N (%) | PRISMA, mean (SD) | AMSTAR, mean (SD) |
|---|---|---|---|
| Renal transplantation | 1 (1%) | 22 | 7 |
| Male infertility | 6 (3%) | 20.5 (1.05) | 6.33 (1.03) |
| Calculi | 39 (21%) | 22.69 (1.98) | 7.49 (1.25) |
| Female urology | 5 (3%) | 23.4 (1.14) | 8.8 (1.09) |
| Neurourology | 42 (23%) | 22.67 (1.72) | 7.67 (1.16) |
| Cannot classify | 2 (1%) | 23 (0) | 7.5 (0.71) |
| **Funding support** | | | |
| Yes | 55 (30%) | 23.31 (2.04)[*] | 7.82 (1.28) |
| No | 128 (70%) | 22.50 (2.00) | 7.46 (1.46) |
| **Number of included studies** | | | |
| ≥10 | 101 (55%) | 22.89 (2.08) | 7.50 (1.46) |
| <10 | 82 (45%) | 22.56 (2.00) | 7.66 (1.35) |
| **Included studies type** | | | |
| Only RCTs | 62 (34%) | 22.82 (1.92) | 7.95 (1.27) |
| RCTs & non-RCTs | 50 (27%) | 22.90 (1.88) | 7.62 (1.40) |
| Only non-RCTs | 71 (39%) | 22.56 (2.27) | 7.20 (1.46) |
| **Included studies type** | | | |
| Only RCTs | 62 (34%) | 22.82 (1.92) | 7.95 (1.27)[**] |
| RCTs & non-RCTs or only non-RCTs | 121 (66%) | 22.70 (2.11) | 7.37 (1.44) |
| **Meta-analyses type** | | | |
| Interventional | 141 (77%) | 22.60 (1.96) | 7.64 (1.37) |
| Diagnostic | 12 (7%) | 23.67 (2.15) | 7.83 (1.11) |
| Incidence | 15 (8%) | 23.40 (2.20) | 7.07 (1.71) |
| Prognostic | 15 (8%) | 22.73 (2.46) | 7.2 (1.61) |
| **Surgical intervention** | | | |
| Yes | 96 (68%[b]) | 22.46 (1.86) | 7.56 (1.36) |
| No | 45 (32%[b]) | 22.89 (2.14) | 7.80(1.41) |
| **PRSIMA checklist** | | | |
| Yes | 2 (1%) | 22.5 (0.71) | 7.5 (0.71) |
| No | 181 (99%) | 22.75 (2.06) | 7.57 (1.42) |
| **Followed PRSIMA** | | | |
| Yes | 55 (30%) | 23.38 (2.13)[**] | 7.94 (1.48)[*] |
| No | 128 (70%) | 22.47 (1.95) | 7.41 (1.35) |
| **Protocol and registration** | | | |
| Yes | 8 (4%) | 26 (1.19)[***] | 9.75 (0.71)[***] |
| No | 175 (96%) | 22.59 (1.95) | 7.47 (1.36) |
| **"*a priori*" design** | | | |
| Yes | 22 (12%) | 24.09 (2.07)[***] | 9.14 (1.13)[***] |
| No | 161 (88%) | 22.56 (1.98) | 7.35 (1.31) |

**Notes.**

[*]$P < 0.05$.
[**]$P < 0.01$.
[***]$P < 0.001$.
[a]81 from mainland China, 1 from Taiwan.
[b]Out of 141.
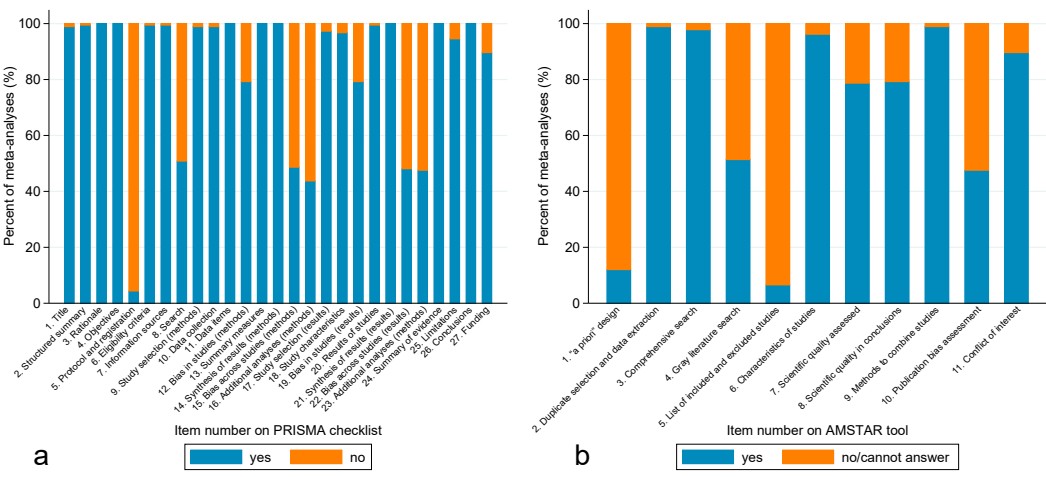

**Figure 2** (A) Bar graph of percentage of meta-analyses that included each item from the PRISMA checklist. (B) Bar graph of percentage of meta-analyses that included each item from the AMSTAR tool.

significant difference of AMSTAR score between the meta-analyses from China and those from remaining countries, $t(181) = -0.25$, $p = 0.80$. Unlike PRISMA score, the AMSTAR score of included meta-analyses in the subspecialty of urologic oncology did not differ from other subspecialties, $t(181) = -0.68$, $p = 0.50$. There was a significant difference of AMSTAR score between the included studies type (Only RCT vs. RCT & non-RCT vs. Only non-RCT), $F(2, 180) = 4.98$, $p = 0.008$. Tukey's HSD *post hoc* test showed that meta-analyses included only RCTs had higher AMSTAR score than those included only non-RCTs, $p = 0.002$. There was no significant difference of AMSTAR score between the interventional and non-interventional meta-analyses, $t(181) = -1.23$, $p = 0.22$. All the other two-group comparison test results are shown in Table 1.

Figure 2B shows the AMSTAR results on a per-item basis. Per-item AMSTAR analysis revealed that 3 items had less than 50% adherence out of the 183 included meta-analyses (item 1, "*a priori*" design; item 5, list of studies; item 10, publication bias). Item 4 (gray literature) also had only 51% adherence.

## Univariate and multivariate analyses

The 75% percentile of the respective ranges of PRISMA score and AMSTAR score was 24 and 9, respectively. The reporting quality (PRISMA) and methodological quality (AMSTAR) were divided by the cutoff value of 75% percentile into superior quality and inferior quality. The results of univariate and multivariate logistic regression analyses on the PRISMA and AMSTAR scores are presented in Tables 2 and 3, respectively.

Univariate regression analyses demonstrated the following factors to be associated with superior reporting quality (PRISMA score ≥ 24) of the published meta-analyses: subspecialty of urologic oncology, funding support, non-interventional meta-analyses, following PRISMA guideline, and "*a priori*" design. Multivariate regression analyses confirmed the following factors to be associated with superior reporting quality (PRISMA score ≥ 24) of the published meta-analyses: funding support and "*a priori*" design.

**Table 2  Univariate and multivariate logistic regression analyses of predictive factors associated with superior reporting quality (PRISMA).**

| Variables | Univariate | | Multivariate | |
|---|---|---|---|---|
| | OR (95%CI) | *P* value | OR (95%CI) | *P* value |
| Asia (ref. non-Asia) | 1.09 (0.59, 2.00) | 0.79 | | |
| China (ref. non-China) | 1.02 (0.55, 1.89) | 0.95 | | |
| No. of authors (continues) | 1.02 (0.92, 1.14) | 0.67 | | |
| No. of authors ≥ 7 (ref. < 7) | 1.55 (0.84, 2.88) | 0.16 | | |
| Presence of epi/stats professional (ref. absence) | 0.77 (0.37, 1.61) | 0.49 | | |
| No. of participating center (continues) | 0.98 (0.87, 1.10) | 0.72 | | |
| Multiple center (ref. single center) | 1.23 (0.60, 2.53) | 0.57 | | |
| Urologic oncology (ref. other subspecialties) | **2.14 (1.14, 3.99)** | **0.02** | 1.41 (0.68, 2.91) | 0.36 |
| Funding support (ref. no) | **2.29 (1.19, 4.41)** | **0.01** | **2.33 (1.16, 4.69)** | **0.02** |
| No. of included studies (continues) | 1.00 (0.99, 1.01) | 0.91 | | |
| No. of included studies ≥ 10 (ref. < 10) | 1.31 (0.71, 2.45) | 0.38 | | |
| Included only RCTs (ref. RCTs & non-RCTs or only non-RCTs) | 1.00 (0.52, 1.91) | 1.00 | | |
| Interventional (ref. non-interventional) | **0.47 (0.23, 0.94)** | **0.03** | 0.45 (0.19, 1.03) | 0.06 |
| Surgical intervention (ref. non-surgical intervention) | 0.58 (0.27, 1.23) | 0.16 | | |
| Followed PRSIMA (ref. no) | **2.04 (1.07, 3.94)** | **0.03** | 1.88 (0.92, 3.87) | 0.09 |
| "*a priori*" design (ref. no) | **3.30 (1.32, 8.23)** | **0.01** | **3.74 (1.40, 10.02)** | **0.01** |

**Notes.**

Significant results are shown in bold.

**Table 3  Univariate and multivariate logistic regression analyses of predictive factors associated with superior methodological quality (AMSTAR).**

| Variables | Univariate | | Multivariate | |
|---|---|---|---|---|
| | OR (95%CI) | *P* value | OR (95%CI) | *P* value |
| Asia (ref. non-Asia) | 0.77 (0.40, 1.49) | 0.43 | | |
| China (ref. non-China) | 0.75 (0.38, 1.46) | 0.40 | | |
| No. of authors (continues) | **1.15 (1.02, 1.29)** | **0.02** | 1.00 (0.87, 1.16) | 0.95 |
| No. of authors ≥ 7 (ref. < 7) | 1.69 (0.87, 3.31) | 0.12 | | |
| Presence of epi/stats professional (ref. absence) | 1.25 (0.59, 2.64) | 0.57 | | |
| No. of participating center (continues) | 1.11 (1.00, 1.25) | 0.06 | | |
| Multiple center (ref. single center) | 1.63 (0.72, 3.71) | 0.24 | | |
| Urologic oncology (ref. other subspecialties) | **2.07 (1.06, 4.04)** | **0.03** | 2.05 (0.92, 4.58) | 0.08 |
| Funding support (ref. no) | 1.59 (0.79, 3.19) | 0.19 | | |
| No. of included studies (continues) | 1.00 (0.99, 1.01) | 0.85 | | |
| No. of included studies ≥ 10 (ref. < 10) | 1.06 (0.55, 2.06) | 0.86 | | |
| Included only RCTs (ref. RCTs & non-RCTs or only non-RCTs) | 1.78 (0.91, 3.51) | 0.10 | | |
| Interventional (ref. non-interventional) | 1.00 (0.46, 2.19) | 1.00 | | |
| Surgical intervention (ref. non-surgical intervention) | 0.82 (0.37, 1.81) | 0.63 | | |
| Followed PRSIMA (ref. no) | **3.80 (1.90, 7.64)** | **<0.001** | **2.94 (1.33, 6.47)** | **0.01** |
| "*a priori*" design (ref. no) | **19.65 (6.19, 62.29)** | **<0.001** | **17.37 (4.98, 60.56)** | **<0.001** |

**Notes.**

Significant results are shown in bold.

Univariate regression analyses demonstrated the following factors to be associated with superior methodological quality (AMSTAR score ≥ 9) of the published meta-analyses: number of authors, subspecialty of urologic oncology, following PRISMA guideline, and "*a priori*" design. Multivariate regression analyses confirmed the following factors to be associated with superior methodological quality (AMSTAR score ≥ 9) of the published meta-analyses: following PRISMA guideline, and "*a priori*" design.

## DISCUSSION

Our study demonstrates that both reporting and methodological qualities of recently published meta-analyses in major urology journals were generally good. Also, there were no significant variations of the qualities between major urology journals. On average, PRISMA score was 22.74 (84.2%) out of 27 and AMSTAR score was 7.57 (68.8%) out of 11. However, there still may be room for improvement based on the per-item results (Fig. 2). More importantly, several potential predictive factors for superior quality of urological meta-analyses were identified, including funding support, following PRISMA guideline, and "*a priori*" design. Knowledge and identification of variables predictive of high-quality meta-analysis are not only useful to readers, but also would be useful for journal reviewers and editors.

Reporting and methodological qualities of meta-analyses in other medical disciplines were evaluated with similar methods (*Liu et al., 2017*; *Zhang et al., 2016*). *Zhang et al. (2016)* focused on meta-analyses of surgical interventions in year 2013 and showed the mean PRISMA and AMSTAR adherences (by items) were 22.3 and 7.9, respectively. A recent study showed the mean PRISMA and AMSTAR adherences (by items) in the leading gastroenterology and hepatology journals were 20.8 and 7.6, respectively (*Liu et al., 2017*). Generally speaking, the quality of meta-analyses in major urology journals are good and consistent with previous studies in other medical fields.

Strengths of our study include the focused search and selection of meta-analyses, comprehensive assessment, and planned logistic regression analyses. We only included meta-analyses because they are different from qualitative systematic reviews in several ways and often have a more consistent format. In addition, some of the assessment items only applied to meta-analyses, such as PRISMA items 14, 15, 16, 21, 22, 23, and AMSTAR items 9 and 10. By excluding systematic reviews without meta-analyses, our results would have more credibility and our conclusions would have a more specific implication. Also, we only focused on meta-analyses published starting from 2011, which is one year and a half after the publication of PRISMA statement (July 2009) (*Moher et al., 2009*). Since the AMSTAR tool was first published in 2007, this timeline setting would possibly minimize the confounding from authors' unavailability of PRISMA checklist and AMSTAR tool themselves.

There are some limitations to our study. First, the cumulative scores calculated from PRISMA checklist and AMSTAR tool may not be valid or truly reflect the reporting and methodological quality. However, at least for now, the scoring method seems to be the best option to quantify the quality of and meta-analyses (*Adie et al., 2015*; *Gagnier & Kellam, 2013*; *Tunis et al., 2013*; *Zhang et al., 2016*). Second, we limited our search to 10 predetermined major paper-based urology journals, which could cause the omitting of
some urological meta-analyses published in general medical journals. Our selection of journals was somewhat arbitrary, even though we took the equal distribution of general urology journals (BJUI, EU, JU, URO, and WJU) and subspecialty journals (JEU, JPU, NUUD, UL, and UO) into consideration.

Reporting and methodical weakness of urological meta-analyses were identified through per-item analyses. PRISMA item 15, 22 and AMSTAR item 10 is about publication bias, one of the most important reporting biases in meta-analyses (*Sedgwick, 2015*). Ideally, for meta-analyses with more than 10 included studies, funnel plots and tests for funnel plot asymmetry should be provided to explore the publication biases (*Sterne et al., 2011*). PRISMA items 16 and 23 (additional analyses) represent another important area for improvement. Less than 50% percent meta-analyses conducted additional analysis, such as subgroup analysis, meta-regression, and sensitivity analysis. Meta-analyses often have the intrinsic limitation of heterogeneity and conclusions could be misleading because of this. Most of the meta-analyses did report the quantified heterogeneity using $I^2$ value or other tests, the source of heterogeneity was not routinely explored. Subgroup analysis and meta-regression can be performed to explain the source of the significant heterogeneity (*Phan et al., 2015*; *Thompson, 1994*). In addition, sensitivity analyses is the method of choice to explore the robustness of the meta-analysis results. PRISMA item 8 (search) and AMSTAR item 5 (list of studies) were also underreported possibly because of the authors' unawareness. Underreported meta-analyses gave partial search strategies such as keywords used or MeSH terms. However, failing to provide detailed or exact search strategy makes it difficult to repeat the search process. Similarly, AMSTAR item 5 (list of studies) were underreported because authors only considered the lists of included studies and neglected the lists of important excluded studies. Suboptimal compliance with AMSTAR item 4 (gray literature) should also be noted since exclusion of gray literature from meta-analyses can lead to exaggerated estimates of intervention effectiveness (*McAuley et al., 2000*). In theory, meta-analyses should attempt to identify and retrieve all potentially eligible studies, including gray literature and foreign language literature (*McAuley et al., 2000*; *Phan et al., 2015*). In reality, however, considerable additional resources are required, which makes it difficult to finish the process.

Multivariate logistic regression showed that claiming a meta-analysis followed the PRISMA statement in the article could potentially predict higher methodological quality (AMSTAR). Although PRISMA guideline is intended to define how to report meta-analyses, it can also be used to design meta-analyses in some way. One worrisome fact is that only 55 of the 183 included meta-analyses claimed followed the PRISMA statement (Table 1). Although it is possible that some of the authors may not have described it in the meta-analysis even if they referred to the PRISMA statement, it is also likely that many authors lacked the knowledge of the availability of PRISMA checklist. Two other facts are that only one journal (EU) endorses the compliance with the PRISMA statement in the instructions for authors and none of the journals required a PRISMA checklist for the initial manuscript submission. Therefore requiring a mandatory PRISMA checklist during submission might improve reporting and methodological quality of meta-analyses. It might be easier to initiate the process from the journals than the authors since only two of the 183 included

meta-analyses attached the PRISMA checklist (*Tewari et al., 2012*; *Van Die et al., 2014*). In addition, having a PRISMA checklist makes the peer review process more efficient and more informed.

Another important predictor of high-quality meta-analyses is "*a priori*" design. As one of the AMSTAR items, "*a priori*" design can predict both the reporting quality and methodological quality. In PRISMA checklist, item 5 is "protocol and registration", which can be considered as the higher standard of "*a priori*" design. It is not hard to understand that "*a priori*" design can make sure the researchers have a clear thinking and well-organized action. In addition, having a protocol or "*a priori*" design can partially obligate the authors from *post hoc* modification of inclusion criteria and analytic methods (*Tunis et al., 2013*). However, only 8 meta-analyses fulfilled the PRISMA item 5 and only 22 meta-analyses claimed "*a priori*" design. In the medical publication, requiring the protocol and registration information for RCTs is very common. As for systematic reviews, only Cochrane reviews require the authors to publish a peer-reviewed protocol before conducting the review. Previous studies have shown that Cochrane reviews appear to have higher methodological quality than systematic reviews or meta-analyses published in paper-based journals. Another common registration platform for systematic reviews and meta-analyses is PROSPERO (*Booth et al., 2012*). It would be very difficult for paper-based journals ask for prospective registration or peer-reviewed protocol for every meta-analysis. Attaching a study protocol written "*a priori*" might be a good start (*Reeves et al., 2015*).

## CONCLUSIONS

Reporting and methodological qualities of recently published meta-analyses in major urology journals are generally good, however there are areas for potential improvement. Further improvement could potentially be achieved by strictly adhering to PRISMA guideline and preparing "*a priori*" protocol.

### Funding

This work was supported by The Linda and Joel Appel Urologic Oncology Research Fund and The Honickman Family Urologic Research Fund. The funders had no role in study design, data collection and analysis, decision to publish, or preparation of the manuscript.

### Grant Disclosures

The following grant information was disclosed by the authors:
The Linda and Joel Appel Urologic Oncology Research Fund.
The Honickman Family Urologic Research Fund.

### Competing Interests

The authors declare there are no competing interests.
## Author Contributions

- Leilei Xia and Jing Xu conceived and designed the experiments, performed the experiments, analyzed the data, contributed reagents/materials/analysis tools, wrote the paper, prepared figures and/or tables, reviewed drafts of the paper.
- Thomas J. Guzzo conceived and designed the experiments, performed the experiments, analyzed the data, contributed reagents/materials/analysis tools, wrote the paper, reviewed drafts of the paper.

## Data Availability

The raw data has been supplied as Data S1.

## Supplemental Information

Supplemental information for this article can be found online at http://dx.doi.org/10.7717/peerj.3129#supplemental-information.

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
