# Peer review of "Reporting and methodological quality of meta-analyses in urological literature"

_PeerJ, doi:10.7717/peerj.3129_

## Round 0.1 · original submission · Major Revisions

Dear authors,

Thank you very much for the opportunity to handle this paper. After reading it with the comments of the reviewers, I think it has scientific merit to be published in PeerJ, but only once the various issues raised by the reviewers are solved by the authors. Therefore, my decision is MAJOR REVISION.

With respect and warm regards,
Dr Palazón-Bru (academic editor for PeerJ)

·

Basic reporting

1. Authors mixed the concepts of SRs and MAs. In different paragraphs, they referred to different objections (SRs or MAs).
2. The labels of Figure 2 are not clear. You can refer to "Quality of Meta-analyses in Major Leading Gastroenterology and Hepatology Journals: A Systematic Review, J Gastroenterol Hepatol. 2016 Sep 6. doi: 10.1111/jgh.13591."

Experimental design

1. The author did not report how they determined the journals. By impact factors or the number of publication?
2. The inter-observer agreement between the two reviewers should be caculated.

Validity of the findings

1. Authors concluded that "Reporting and methodological qualities of recently published meta-analyses in major urology journals are generally good", but the conclusion was not strongly supported by results or discussions. It would be better if conpare with the quality in other disciplines.
2. In fact, it is stated that PRISMA is not quality assessment instrument to gauge the quality of a systematic review, so authors should discuss it.
3. Please shorten the part of strength.

Reviewer 2 ·

Basic reporting

No comments

Experimental design

No comments

Validity of the findings

This is an interesting paper but there are some of major considerations, if addressed I believe they would really enhance this manuscript:

1. The most important flaw of this manuscript concern the search strategy and the inclusion and exclusion criteria that may lead to a selection bias. As stated by the authors, limiting the search strategy to some urology journals is a limit of the study. A search on Pubmed must be repeated without any limitation of journals and its results compared with the results currently reported in the manuscript.

2. Moreover, the authors must explain why they decide to exclude network and multiple group meta-analyses and, most important thing, why they did not include meta-analyses published in the Cochrane Database of Systematic Review. I strongly advise to carry out a specific search on this database.

3. The statistical analysis should be improved including also or only the results of non-parametric tests. To apply parametric tests one would assume that the AMSTAR and PRISMA scores are normally distributed. A specific test for the normality must be carried out and then the variables must be compared with the appropriate tests.

4. Another important flaw of the study is the selection of the covariate to be included in the multiple logistic regression. I was wondering why the authors included the “a priori” design as a covariate. One would include in the model only variables that are not included in the AMSTAR and that do not contribute to the final score used as outcome.

Additional comments

The second paragraph of the Introduction section must be shortened. An additional paragraph, describing the studies already carried out using the same methodology to assess the methodological quality of meta-analyses in other fields of research, must be included in this section.

I hope you find the comments helpful for re-drafting as there is lots of interesting work in the manuscript.

·

Basic reporting

Introduction section

1) The paragraph 56 to 59 is suggested to revise as something like “The systematic review may include a quantitative synthesis of results called meta-analysis. The meta-analysis summarizes all results of primary studies in order to obtain a combined estimate of the treatment effect. Systematic review and meta-analysis are considered as the highest level of evidence.”

2) Please added a reference line 59 to 60 “also, well-conducted meta-analyses can….more reliable conclusion”

Methods section
1) Suggest moving eligibility criteria before search strategy

2) I suggest moving the paragraph “data extraction” before the paragraph line 122 to 145. The authors should summarise all data collected for example: “We collected all data on general characteristics of systematic reviews and meta-analyses, the key methodological components of the systematic review process. Two reviewer independently extracted data. Disagreements were resolved by discussion.”
After the paragraph data extraction, the authors may include a new paragraph line 151 titled “General characteristic of systematic review and meta-analyses”.


3) Line (148) the authors should clarify how disagreement were managed during data extraction.


4) I suggest moving the paragraph line 122 to 145 in new paragraph titled “Assessment of key methodological components of the systematic review process”. The authors may mention each items assessed for example: “We assessed whether the report described how many people performed the study selection and the data extraction and if they used a consensus procedure for disagreements. We evaluated whether the methodological quality or risk of bias of primary studies was assessed and how”
“We assessed whether the results of primary studies and selection process were documented, including the number of studies screened, assessed for eligibility, and included in the review, with reasons for exclusions at each stage”

Results section

1) Figure1: Need to revise this figure. Base on recommendation of PRISMA, the authors may provide numbers of studies screened, assessed for eligibility, and included in the review, with reasons for exclusions AT EACH STAGE, ideally in their flow diagram.

Experimental design

Methods section

1) In the methods section the authors may provide explanations on why they exclude “Epud ahead of print” (113) and meta-analysis of single proportions (line 117), it is unclear to me especially


2) The authors should mention the number of patients included per meta-analysis

3) More detailed on type of funding (private or public fund) and type of intervention should be mentioned for example, how many MA assessed surgical intervention, pharmacological intervention, educational intervention, procedure (device). It will be more interesting



4) PRISMA focusing on reporting and AMSTAR on a mixture of reporting and methods. Authors may differentiate all items related to reporting from methodology in AMSTAR for assessment of methodological quality
5) When reading through the manuscript, I was confused on the difference between reporting and conduct/performance quality. The fact that some of the conclusions on conduct quality are based on the assessment of reporting quality may point to the fact that the authors actually mixed these two concepts (when the author pooled items were rated “no” and “can’t answer.) However, there is empirical evidence from primary clinical studies that these are actually two different things and the authors need to be careful to disentangle this (e.g. Mhaskar R et al. Published methodological quality of randomized controlled trials does not reflect the actual quality assessed in protocols. J Clin Epidemiol 2012; 65:602-9). In general, I think the authors are able to assess methodological quality only if the reporting was adequate in published reports.
To clarify this difference, the author should proceeding in two steps: 1) assess whether items were reported; and 2) if adequately reported, assess whether the conduct quality was adequate. For example, if the authors did not report which databases were used for their search, we considered the reporting of this item inadequate; when the authors reported that they searched for all published RCTs in only one bibliographic database (e.g., MEDLINE), the authors should considered the reporting adequate, and conduct quality as inadequate. Because since they have no reason to believe that the authors of reviews really searched more than one bibliographic database (a search should involve at least 2 electronic databases.)

.
6)I am not convinced that use of PRISMA and AMSTAR score was pertaining. The authors considered that each item of PRISMA or AMSTAR were equivalent and same importance for assessment of methodological quality. In AMSTAR, the instruments include items that are more closely related to reporting quality rather than to the internal validity of systematic review. For example, AMSTAR assessed whether an “a priori” design was provided, whether the methods used to combine the finding of studies was appropriate. I suggest identifying, assessing and presenting individually each item specific of reporting and methodological aspects.

7) In paragraph data analysis please describe what factors were selected for univariate and multivariate logistic regression analysis

Results section

2) Figure 2 and 3 is not pertaining

Validity of the findings

no comment

Additional comments

Thank you for the opportunity to review the manuscript entitled “Reporting and methodological quality of meta-analyses in urological literature”. The issue is interesting for readers. However, the paper would benefit from a major editorial review.
1) Distinction between reporting and conduct quality to be made more clearly
2) Use of AMSTAR and PRISMA score was not pertaining for assessment of methodological aspect. It is will be more interesting to identify what items were not reported rather quality score

---

## Round 0.2 · accepted · Accept

Dear authors,

Thank you very much for the opportunity to handle this paper. All the comments of the three reviewers have been addressed correctly. Therefore, your manuscript has high standards to be published in PeerJ in its current form. So, my decision is TO ACCEPT.

Congratulations!

With respect and warm regards,
Dr Palazón-Bru (academic editor for PeerJ)

·

Basic reporting

Much better.

Experimental design

The authors revised all the questions.

Validity of the findings

Much stronger.

Reviewer 2 ·

Basic reporting

None

Experimental design

None

Validity of the findings

None

Additional comments

The authors have satisfactorily responded to all my questions and made the necessary changes to the manuscript. It looks ready for publication as far as I can tell.